Improving maize’s N uptake and N use efficiency by strengthening roots’ absorption capacity when intercropped with legumes

Zheng Benchuan 1 2
Zhang Xiaona 1 2
Chen Ping 1 2
Du Qing 1 2
Zhou Ying 1 2
Yang Huan 1 2
Wang Xiaochun 1 2
Yang Feng 1 2
Yong Taiwen 1 2 yongtaiwen@sicau.edu.cn
Yang Wenyu 1 2
1 College of Agronomy, Sichuan Agricultural University , Chengdu , China
2 Sichuan Engineering Research Center for Crop Strip Intercropping System/Key Laboratory of Crop Ecophysiology and Farming System in Southwest, Ministry of Agriculture , Chengdu , China
Kalaji Hazem
Electronic publication date: 2021 Jun 23
Publication date: 2021
Volume: 9
Electronic Location ID: e11658
Received 2021 Feb 19; Accepted 2021 Jun 1
Copyright: © 2021 Zheng et al.
Copyright year: 2021
Copyright holder: Zheng et al.
License: This is an open access article distributed under the terms of the Creative Commons Attribution License, which permits unrestricted use, distribution, reproduction and adaptation in any medium and for any purpose provided that it is properly attributed. For attribution, the original author(s), title, publication source (PeerJ) and either DOI or URL of the article must be cited.
License URL: https://creativecommons.org/licenses/by/4.0/

Keywords: Maize-soybean strip intercropping, Maize-peanut strip intercropping, Nitrogen, Root distribution, Antioxidant enzyme activity, Root bleeding sap intensity

Funding: National Natural Science Foundation of China 31872856 National Key Research and Development Program of China 2016YFD030020205 This work was supported by the National Natural Science Foundation of China (31872856) and the National Key Research and Development Program of China (2016YFD030020205). The funders had no role in study design, data collection and analysis, decision to publish, or preparation of the manuscript.

==============================
Maize’s nitrogen (N) uptake can be improved through maize-legume intercropping. N uptake mechanisms require further study to better understand how legumes affect root growth and to determine maize’s absorptive capacity in maize-legume intercropping. We conducted a two-year field experiment with two N treatments (zero N (N0) and conventional N (N1)) and three planting patterns (monoculture maize (Zea mays L.) (MM), maize-soybean (Glycine max L. Merr.) strip intercropping (IMS), and maize-peanut (Arachis hypogaea L.) strip intercropping (IMP)). We sought to understand maize’s N uptake mechanisms by investigating root growth and distribution, root uptake capacity, antioxidant enzyme activity, and the antioxidant content in different maize-legume strip intercropping systems. Our results showed that on average, the N uptake of maize was significantly greater by 52.5% in IMS and by 62.4% in IMP than that in MM. The average agronomic efficiency (AE) of maize was increased by 110.5 % in IMS and by 163.4 % in IMP, compared to MM. The apparent recovery efficiency (RE) of maize was increased by 22.3% in IMS. The roots of intercropped maize were extended into soybean and peanut stands underneath the space and even between the inter-rows of legume, resulting in significantly increased root surface area density (RSAD) and total root biomass. The root-bleeding sap intensity of maize was significantly increased by 22.7–49.3% in IMS and 37.9–66.7% in IMP, compared with the MM. The nitrate-N content of maize bleeding sap was significantly greater in IMS and IMP than in MM during the 2018 crop season. The glutathione (GSH) content, superoxide dismutase (SOD), and catalase (CAT) activities in the root significantly increased in IMS and IMP compared to MM. Strip intercropping using legumes increases maize’s aboveground N uptake by promoting root growth and spatial distribution, delaying root senescence, and strengthening root uptake capacity.

Introduction

Intercropping produces higher crop yields (Waghmaref & Singh, 1984; Li et al., 2001; Beedy et al., 2010), increases nutrients, and water use efficiency (Rahman et al., 2016; Yong et al., 2018), reduces the need for fertilizers (Liu et al., 2014; Yong et al., 2014; Luo et al., 2016), and maintains soil fertility (Wang et al., 2015). The cereal-legume intercropping system has attracted attention in recent years due to legumes’ symbiotic nitrogen fixation (Jiao, 2008; Zhang et al., 2017).

The root system absorbs and utilizes soil water and nutrients, which promotes root growth. Crops may improve soil nutrient absorption through root proliferation in nutrient-enriched regions (Chilundo et al., 2017). However, root growth and development also affects the soil nutrient cycle and nutrient availability through root exudates (Li et al., 2007; Coskun et al., 2017; Meier, Finzi & Phillips, 2017). The organic acids in root exudates activate soil nutrients (Li et al., 2007) and modify soil microbial community structure (Baudoin, Benizri & Guckert, 2003; Haichar et al., 2008; Badri & Vivanco, 2009). Crops can adapt to the non-uniform distribution of mineral nutrients in soil through root plasticity (Yu et al., 2014). Root length density (RLD), root weight density (RWD), and root surface area density (RSAD) can be used to quantify crop root extension and distribution (Liu et al., 2020; Ren et al., 2017). Root-bleeding sap is also an important indicator of root activity as the components of the sap reflect the root system’s ability to uptake and transport substances (Guan et al., 2014; Jia et al., 2018). Previous studies have determined that intercropping can promote root growth and modify root distribution (Gao et al., 2010; Ren et al., 2017; Liu et al., 2020). In one study, maize roots extended into soybean rows and maize RLD increased in the topsoil layer, while soybean roots were mainly located near the plants (Gao et al., 2010). In another study, maize in a wheat-maize intercropping system modified its root distribution and RLD to increase N uptake per unit root length in an area occupied by wheat crops (Liu et al., 2020). Few studies have been conducted on maize roots in a maize-peanut strip intercropping system. In addition to root distribution, root bleeding sap intensity is an important indicator of root activity, and the components of bleeding sap reflect the nutrients of root absorption and transport (Yang et al., 2016; Zhang et al., 2007a). Intercropping may affect crops’ root bleeding intensity. Planting patterns and maize row spacing decreased root bleeding in soybeans, which then influenced the nutrient uptake in this maize-soybean relay strip intercropping system (Yang et al., 2016).

Changes in root antioxidation are an emergency response for crops needing to adapt to variations in the soil environment, e.g., water (Hu et al., 2010), nutrient (Liu & Jiang, 2017; Yao et al., 2019), heavy metal (Maiti et al., 2012; Zhang et al., 2007b), and salt stress (Zhu et al., 2004; Shalata & Tal, 2010). Superoxide anion radicals (O2−) and hydrogen peroxide (H2O2) are induced when plants suffer from environmental stress (Bowler et al., 2011; Maiti et al., 2012). Reactive oxygen species (ROS) are toxic for the growth and development of plants and antioxidant enzymes. Antioxidants, including superoxide dismutase (SOD), catalase (CAT) and glutathione (GSH), help eliminate the excess ROS and maintain the intracellular homeostasis (Gill & Tuteja, 2010). If root antioxidation responds quickly, then the soil environment may delay root senescence (Hu et al., 2010; Mucha et al., 2012). However, few studies have been conducted on delaying senescence in a root system belonging to a maize-legume strip intercropping system.

Maize-soybean strip intercropping (IMS) and maize-peanut strip intercropping (IMP) are two popular planting patterns used in Chinese agriculture. Maize has a greater N uptake when intercropped with legumes (Zhang et al., 2017), which may be the result of belowground interactions, such as root interactions, interspecific facilitation, and the competitive use of nutrients (Li et al., 2001; Xia et al., 2013; Liu et al., 2020). Previous studies have shown that intercropping can increase nutrient uptake by altering root plasticity (Gao et al., 2010; Xia et al., 2013; Ren et al., 2017; Liu et al., 2020). However, it is still unclear what impact intercropping legumes with other plants has on root growth and maize. Additionally, delayed root senescence and the influence of legumes on antioxidants (e.g., enzyme activity) in the maize root system still needs further study. Therefore, we hypothesized that maize intercropped with legumes will increase N uptake by improving the root’s spatial distribution by expanding the nutrient acquisition area, enhancing maize roots’ antioxidant capacity to delay root senescence and increase the nutrient acquisition time, and increasing the root bleeding intensity to strengthen roots’ nutrient acquisition ability. The objective of this study was to clarify the influence of legumes on the root growth and maize’s nutrient use in maize-legume strip intercropping systems. With this aim, we studied the RSAD, root biomass, root bleeding sap intensity, root antioxidant enzyme activity, and root antioxidants of maize.

Materials and Methods

Experiment site

Our field experiment was performed in Renshou County (30°16′N, 104°00′E), Sichuan Province, Southwest China, from April to November during the 2017 and 2018 crop seasons. The experimental site has a subtropical monsoon humid climate with an annual temperature of 17.4 °C and annual precipitation of 1,009.4 mm. The temperature and precipitation during the cropping seasons are shown in Fig. 1. The soil is anthrosol with a clay loam texture and the nutritional characteristics of the topsoil are as follows: 14.19 g kg−1 of organic matter, 1.22 g kg−1 of total N, 1.95 g kg−1 of total P, 26.06 g kg−1 of total K, and an average pH of 8.18.

Figure 1 Precipitation and temperature during the cropping season in 2017 and 2018.

Experimental design and crop management

We designed a split-plot experiment with three replicates. The main variable was N application rates with no N fertilizer (N0) and conventional N fertilizer (N1); the sub-factor was planting patterns, including monoculture maize (MM), monoculture soybean (MS), monoculture peanut (MP), maize-soybean strip intercropping (IMS), and maize-peanut strip intercropping (IMP). The plots measured 5.8 × 6.0 m. Crop density was 100,000 plants ha−1 for MM and 200,000 plants ha−1 for both MS and MP. Rows were spaced 0.5 m apart in all three types. In MM, plants were spaced 0.2 m apart and for monoculture legumes (MS and MP) plants were spaced 0.1 m apart. Two rows of maize were replaced by two rows of legumes in the two maize-legume strip intercropping systems. Spacing between plants was the same as the corresponding monocultures. Crop density was 50,000 plants ha−1 for maize and 10,000 plants ha−1 for legumes. The conventional N rate (N1) was 240 kg N ha−1 for MM and 80 kg N ha−1 for both MS and MP. The amount of N applied in each intercropping system depended on the proportion of crops compared to the corresponding monocultures. The total N rate was 120 kg N ha−1 for intercropped maize (IM) and 40 kg N ha−1 for intercropped legumes (MS and MP). P and K fertilizers were applied at 120 kg P2O5 ha−1 and 100 kg K2O ha−1 in all planting patterns. We used the maize cultivar “Xianyu-335”, the soybean cultivar “Nandou-12”, and the peanut cultivar “Tianfu-18”. Crops were sown and harvested by artificially. In the 2017 planting year, maize was sown on April 8 and harvested on August 4, soybean was sown on June 9 and harvested on November 1, and peanut was sown on April 7 and harvested on September 13. In the 2018 planting year, maize was sown on April 5 and harvested on August 1, soybean was sown on June 5 and harvested on November 5, and peanut was sown on April 7 and harvested on September 10.

Root growth, antioxidant enzyme activity and antioxidants content investigation

Maize root samples were collected at the silking stage. We collected three soil cores (P1-P3) from the maize monoculture (Fig. 2A) and five soil cores (P1-P5) from intercropped maize (Fig. 2C) to determine the roots’ spatial distribution. The soil cores were collected using a 10 cm auger at the base of the maize plant and 25 cm away. Soil cores were collected at 20 cm intervals to a maximum depth of 100 cm. Maize roots were scanned at a 300 dpi resolution (Epson expression 10000 XL (Japanese) Co., Ltd). The scanned root images were analyzed using Win-RHIZO™ software (Régent Instruments Inc., Canada).

Figure 2 Planting patterns and root sampling sites in the field experiment.

(A) Monoculture maize (MM); (B) monoculture legume (ML); (C) intercropped maize (IM). P1, the inter-row of maize; P2, intra-row of maize; P3, adjacent row of maize and legume; P4, intra-row of legume; P5, inter-row of legume.

We collected roots from six maize plants using a traditional excavation method to obtain 0.20 × 0.50 × 0.30 m soil clods. These samples were used to calculate the total root biomass of a single plant and determine antioxidant enzymes activities and the antioxidant content of maize. The root samples were washed in ice water. Three roots samples were dried at 85 °C to a constant weight. Three plant roots samples were stored in liquid nitrogen and taken to the laboratory where they were stored at −80 °C for further investigation. SOD activity was determined using the nitrogen blue tetrazole (NBT) method at 560 nm (Li et al., 2019). The CAT activity was determined by measuring the abs decrease at 30 s intervals at 240 nm (Zhang et al., 2018). The GSH content was measured using the DTNB method (5,5′-dithiobis-2-nitrobenoic acid) at 412 nm (Li, Wang & Luo, 2018). We determined the physiological parameters using prepared kits (Beijing Solarbio Science & Technology Co., Ltd., Beijing: SOD, BC0170; CAT, BC0200; GSH, BC1175).

Root bleeding intensity and nitrate-N content investigation

We collected sap from roots using a modified technique from Guan et al. (2014). Three maize plants were sampled at the twelfth-leaf stage (V12), the silking stage (R1), and the milk stage (R3). Maize plants were cut 3–4 cm from the internode (about 12 cm above the soil surface) at 6:00 pm. Skimmed cotton was put into a self-sealing bag, placed on the maize stalk, and fixed with a rubber band. The sap in the skimmed cotton was collected and weighed after 12 h. The weight by difference method was used to estimate the intensity of the bleeding sap (g plant−112h−1). We determined the nitrate-N content of maize sap using a Cleverchem Anna Random Access Analyzer (DeChem-Tech.GmbH-Hamburg, Germany).

Plant sampling and determination of plant N content

Three plants were sampled at the maturity stage (R6) in each treatment. Plant samples were categorized as stems, leaves, or kernel. Samples were dried at 105 °C for 30 min to kill living plant tissue. Next, they were dried at 85 °C to a constant weight. Samples were ground and passed through a 60-mesh sieve (0.25 mm). We determined the total N-content using a Cleverchem Anna Random Access Analyzer (DeChem-Tech.GmbH-Hamburg, Germany). The N-content was measured using the sulfuric acid-sodium salicylate method.

Calculations:

Maize N uptake was calculated as follows:

(1) Nuptake(gplant−1)=Nconcentration×Drymatteryield

RSAD was defined as the root surface area per unit soil volume, which was calculated using the following formula:

(2) RSAD=SV

Where RSAD is the root surface area density (cm cm−3), S is the root surface area (cm2), and V is the soil sample volume (1,570 cm3).

We used the agronomic efficiency (AE) and apparent recovery efficiency (RE) to determine maize’s N-use efficiency (NUE) under different planting patterns (Gao et al., 2020). The following equations were used:

(3) AE(kgkg−1)=yieldwithNapplication(kgkg−1)−yieldwithoutNapplication(kgkg−1)totalNapplication(kgkg−1)

(4) RE(%)=TotalNuptakewithNapplication(kgkg−1)−TotalNuptakewithoutNapplication(kgkg−1)TotalNapplication(kgkg−1)×100

Statistical analysis

We used two-way ANOVA analysis to test the influence of N levels and legumes on N uptake and the physiological conditions of different planting patterns. Fisher’s least significant difference (LSD, α = 0.05) was used for data analysis, and our analyses were performed with SPSS v.22 and Microsoft Excel. SigmaPlot14.0 (Systat Software Inc., San Jose, CA, USA), Origin 2017 (OriginLab Corporation, Northampton, MA, USA) and Surfer v. 8.0 (Golden Software LLC, Golden, CO, USA) were used to draw the figures.

Results

N uptake and NUE

Intercropping significantly increased maize’s aboveground N uptake compared with monoculture maize in our two-year field experiment (Table 1). On average, the N uptake of maize stem, leaf, kernel, and total accumulation increased by 27.6%, 35.4%, 63.9% and 52.5% in IMS, respectively, and increased by 53.9%, 42.5%, 68.6% and 62.4% in IMP, respectively, when compared with the MM. The total aboveground N uptake of maize in IMP was 8.4% greater than IMS in 2017. The N application significantly increased the aboveground accumulation of N in maize in all three planting patterns. Planting patterns significantly influenced RE, but there was little effect on AE (Table 2). AE was significantly influenced by the planting year (Table 2). The average maize AE peaked in IMP (7.26 kg kg−1), followed by IMS (6.75 kg kg−1), and MM (4.49 kg kg−1). AE significantly increased by 110.5% in IMS and 163.4% in IMP, compared to MM. RE increased by 22.3% in IMS and decreased by 2.6% in IMP compared with MM.

Table 1 Aboveground N uptake of maize under different N application and planting patterns at the full-maturity stage (g plant−1).

Treatments		Stem	Leaf	Kernel	Total	
		N0	N1	N0	N1	N0	N1	N0	N1	
2017	MM	0.40 ± 0.01 c	0.51 ± 0.01 c	0.22 ± 0.01 b	0.29 ± 0.00 b	1.21 ± 0.01 c	1.55 ± 0.03 b	1.82 ± 0.00 c	2.35 ± 0.04 c	
	IMS	0.50 ± 0.02 b	0.58 ± 0.02 b	0.37 ± 0.04 a	0.40 ± 0.00 a	1.82 ± 0.05 b	2.44 ± 0.03 a	2.69 ± 0.11 b	3.42 ± 0.11 b	
	IMP	0.62 ± 0.02 a	0.70 ± 0.03 a	0.35 ± 0.01 a	0.41 ± 0.02 a	2.10 ± 0.00 a	2.39 ± 0.02 a	3.07 ± 0.01 a	3.50 ± 0.01 a	
2018	MM	0.29 ± 0.02 c	0.44 ± 0.01 b	0.12 ± 0.00 b	0.18 ± 0.00 b	0.73 ± 0.04 b	1.26 ± 0.15 b	1.15 ± 0.05 b	1.88 ± 0.06 b	
	IMS	0.42 ± 0.02 b	0.56 ± 0.03 a	0.16 ± 0.01 a	0.19 ± 0.01 b	1.47 ± 0.04 a	1.87 ± 0.11 a	2.04 ± 0.05 a	2.62 ± 0.11 a	
	IMP	0.54 ± 0.01 a	0.61 ± 0.04 a	0.16 ± 0.02 a	0.25 ± 0.01 a	1.44 ± 0.04 a	1.89 ± 0.17 a	2.14 ± 0.06 a	2.74 ± 0.21 a	
ANOVA (F-value)	
Year (Y)	103.51**		967.26**		432.25**		671.93**		
N application (N)	223.20**		124.36**		368.74**		483.06**		
Planting patterns (P)	290.14**		110.01**		467.23**		578.35**		
Y × N	4.14ns		0.06ns		0.82ns		1.76ns		
Y × P	2.99ns		34.28**		6.74**		8.59**		
N × P	5.82**		8.47**		3.34ns		2.32ns		
Y × N × P	2.48ns		2.02ns		8.45**		4.32*		
Notes:

* Significant difference (P < 0.05) respectively.

** Highly significant difference (P < 0.01) respectively.

ns No significant difference (P > 0.05) respectively.

Different lowercase letters indicate significant differences under different planting patterns in the same cropping seasons (LSD, P < 0.05).

MM, monoculture maize; IMS, maize-soybean strip intercropping system; IMP, maize-peanut strip intercropping system. N0, no N fertilizer; N1, conventional N fertilizer.

Table 2 Agronomic efficiency (AE) and apparent recovery efficiency (RE) of maize as influencd by planting patterns.

	2017	2018	
	AE kg grain kg-1 fertilizer N	RE (%)	AE kg grain kg-1 fertilizer N	RE (%)	
MM	1.75 ± 1.01 b	22.25 ± 1.44 b	7.24 ± 0.44 a	22.24 ± 0.56 a	
IMS	5.41 ± 0.94 a	30.43 ± 4.67 a	8.10 ± 0.78 a	24.00 ± 2.96 a	
IMP	7.53 ± 2.44 a	18.05 ± 1.67 b	6.99 ± 0.19 a	25.27 ± 5.99 a	
ANOVA (F-value)	
Year (Y)	7.10*	0.02ns			
Planting patterns (P)	3.17ns	4.69*			
Y × P	3.32ns	5.86*			
Notes:

* Significant difference (P < 0.05) respectively.

** Highly significant difference (P < 0.01) respectively.

ns No significant difference (P > 0.05) respectively.

Different lowercase letters indicate significant differences under different planting patterns in the same cropping seasons (LSD, P < 0.05).

MM, monoculture maize; IMS, maize-soybean strip intercropping system; IMP, maize-peanut strip intercropping system.

RSAD distribution

The RSAD of monocultured maize indicated that roots had a horizontal symmetrical distribution (Figs. 3A, 3G, 3D and 3J). Roots with an asymmetrical distribution was observed in maize in intercropping systems (Figs. 3B–3C, 3H–3I, 3E–3F and 3K–3L). Maize roots extended into, under (Fig. 2C), and even across the legume inter-rows (Fig. 2C) (i.e., soybean or peanut rows) in the intercropping systems. Maize roots were distributed in the 0–60 cm soil layer. The higher RSAD was observed in the top layers of soil (0–20 cm). Compared with MM, the total RSAD of maize under P2 (Fig. 2C) significantly increased by 21.5% in IMS and by 24.9% in IMP. Intercropped maize’s RSAD was higher at the P3 site than at the P1 site at most soil depths. Maize’s total RSAD was greater by 11.9% in IMP than in IMS under the N1 treatment. Maize’s total RSAD was lower in IMP by 6.4% than in IMS under the N0 treatment. Lastly, maize’s total RSAD increased using the N application in the different planting patterns.

Figure 3 Spatial root surface area density (RSAD) (cm2 cm−3) distribution of maize gume.

MM, monoculture maize; IMS, intercropped maize with soybean; IMP, intercropped maize with peanut. P1, the inter-row of maize; P2, intra-row of maize; P3, adjacent row of maize and legume; P4, intra-row of legume; P5, inter-row of legume. N0, no N fertilizer; N1, conventional N fertilizer.

Root biomass

Maize root biomass was significantly higher in the intercropping system than in the monocultures (Table 3). Under the N0 treatment, the root biomass of maize was significantly increased by 52.6% in IMS and 64.7% in IMP compared with the MM. Under the N1 treatment, maize root biomass significantly increased by 60.4% in IMS and 82.3% in IMP versus MM. Intercropped maize root biomass was higher in IMP than in IMS and significantly increased by 11.8% in IMP compared with IMS in 2018 (Table 3).

Table 3 The total root biomass of maize under different N application rates and planting patterns at the silking stage (g plant−1).

Treatments	N0	N1	
2017	MM	9.90 ± 1.15 b	10.27 ± 0.70 b	
	IMS	15.54 ± 0.32 a	17.98 ± 2.53 a	
	IMP	16.42 ± 1.14 a	20.51 ± 1.18 a	
2018	MM	9.89 ± 0.32 c	11.52 ± 0.11 c	
	IMS	14.66 ± 0.25 b	16.89 ± 0.19 b	
	IMP	16.17 ± 0.68 a	19.13 ± 0.79 a	
ANOVA (F-value)	
Year (Y)	1.34ns		
N application (N)	45.91**		
Planting patterns (P)	187.85**		
Y × N	0.00ns		
Y × P	2.28ns		
N × P	4.66*		
Y × N × P	1.06ns		
Notes:

* Significant difference (P < 0.05) respectively.

** Highly significant difference (P < 0.01) respectively.

ns No significant difference (P > 0.05) respectively.

Different lowercase letters indicate significant differences under different planting patterns in the same cropping seasons (LSD, P < 0.05).

MM, monoculture maize; IMS, maize-soybean intercropping system; IMP, maize-peanut intercropping system. N0, no N fertilizer; N1, conventional N fertilizer.

Root bleeding intensity and nitrate-N content of sap

There were significant differences in maize’s bleeding intensity in monocultured and intercropped systems at different stages of growth. The bleeding intensity of maize significantly increased with the application of N (Table 4). The intensity of the bleeding in maize increased at the R1 stage and then decreased at the R3 stage as the plant grew. Similar trends were observed at the same growth stages in different planting patterns. The intensity of bleeding was significantly enhanced in IMS and IMP compared with the MM, independent of the growth state in maize. Compared with the MM, the bleeding intensity over a two-year average was significantly increased by 49.3%, 28.4% and 22.7% in IMS at the V12, R1, and R3 stages, respectively. The bleeding intensity significantly increased by 66.7%, 40.0% and 37.9% in IMP contrast in MM at the V12, R1 and R3 stages, respectively. Maize’s nitrate-N content significantly increased by 87.4% in IMS and by 96.8% in IMP compared with MM (Fig. 4).

Figure 4 The nitrate-N content of maize bleeding sap under different N application and planting patterns in 2018.

Different lower-case letters indicate significant differences under different planting patterns (LSD, P < 0.05). Vertical bars are standard errors. The asterisk (*) and (ns) indicate significant difference (P < 0.05), highly significant difference (P < 0.01) and no significant difference (P > 0.05), respectively. MM, monoculture maize; IMS, maize-soybean strip intercropping system; IMP, maize-peanut strip intercropping system. N0, no N fertilizer; N1, conventional N fertilizer.

Table 4 The bleeding intensity of maize under different N application and planting patterns (g plant-1 12h-1).

Treatments		V12	R1	R3	
		N0	N1	N0	N1	N0	N1	
2017	MM	8.72 ± 0.32 c	8.91 ± 0.50 c	11.59 ± 0.02 c	14.93 ± 0.24 c	8.50 ± 0.33 b	9.78 ± 0.52 b	
	IMS	15.28 ± 0.97 b	15.82 ± 0.36 b	17.09 ± 0.67 b	18.84 ± 0.50 b	8.60 ± 0.49 b	9.76 ± 0.67 b	
	IMP	17.63 ± 0.96 a	18.18 ± 0.33 a	19.71 ± 0.19 a	22.04 ± 0.65 a	9.65 ± 0.25 a	12.87 ± 0.65 a	
2018	MM	10.60 ± 0.09 c	12.87 ± 0.11 c	12.62 ± 1.00 b	13.92 ± 0.32 b	3.96 ± 0.23 b	5.03 ± 0.89 b	
	IMS	13.57 ± 0.61 b	14.95 ± 0.39 b	15.44 ± 0.50 a	16.36 ± 0.41 a	5.92 ± 0.54 a	7.06 ± 0.18 a	
	IMP	14.48 ± 0.02 a	15.93 ± 0.52 a	15.59 ± 0.17 a	16.50 ± 0.04 a	6.19 ± 0.78 a	7.55 ± 0.40 a	
ANOVA (F-value)	
Year (Y)	4.14**		204.16**		473.07**		
N application (N)	37.37**		119.84**		73.56**		
Planting patterns (P)	465.24**		368.65**		52.45**		
Y × N	13.20**		19.94**		3.77ns		
Y × P	93.68**		76.29**		11.75**		
N × P	0.24ns		3.35ns		4.37*		
Y × N × P	1.34ns		1.18ns		2.62ns		
Notes:

* Significant difference (P < 0.05) respectively.

** Highly significant difference (P < 0.01) respectively.

ns No significant difference (P > 0.05) respectively.

Different lowercase letters indicate significant differences under different planting patterns in the same cropping seasons (LSD, P < 0.05).

MM, monoculture maize; IMS, maize-soybean intercropping system; IMP, maize-peanut intercropping system. N0, no N fertilizer; N1, conventional N fertilizer. V12, the twelve-leaf-stage of maize; R1, the silking-stage of maize; R3, the milk-stage of maize.

Antioxidant enzyme activity and antioxidants of root

The antioxidant enzyme activity and antioxidants of the roots were significantly increased in the intercropping systems compared with the monocultured crops (Fig. 5). The antioxidant capacity of maize was enhanced by N application with different planting patterns (Fig. 5). The SOD activity was significantly increased by 53.2% in IMS and 99.8% in IMP compared with MM (Fig. 5A). The CAT activity was significantly increased by 73.3% in IMS and 113.6% in IMP compared to MM (Fig. 5B). The GSH content was 26.0% higher in IMS and 32.8% in IMP than MM (Fig. 5C). The SOD and CAT enzyme activities were significantly greater by 30.2% and by 23.2% in IMP than in IMS, respectively (Figs. 5A, 5B).

Figure 5 The antioxidant enzyme activity and antioxidant content in maize root under different N application and planting patterns in 2018.

Different lower-case letters indicate significant differences under different planting patterns (LSD, P < 0.05). Vertical bars are standard errors. The asterisk (*) and (**) and (ns) indicate significant difference (P < 0.05), highly significant difference (P < 0.01) and no significant difference (P > 0.05), respectively. MM, monoculture maize; IMS, maize-soybean ralay strip intercropping system; IMP, maize-peanut strip intercropping system. N0, no N fertilizer; N1, conventional N fertilizer.

Discussion

Intercropping increased the nutrient uptake of wheat, soybean, chickpea, and maize. This effect has been reported in numerous studies (Li et al., 2003, 2004; Zhang et al., 2017). We found that maize’s total aboveground N uptake significantly increased by 52.5% in IMS and significantly increased by 62.4% in IMP compared with the MM over 2 years. N uptake was significantly higher by 2.5–14.3% in IMP than in IMS (Table 1). Maize AE was greater by 110.5% in IMS and 163.4% in IMP than in MM over an average of 2 years. Maize RE in IMS increased by 36.8% in 2017 and by 7.9% in 2018 compared with MM. RE in IMP was decreased by 18.9% in 2017 and increased by 13.6% in 2018 compared with MM (Table 2). This effect may be due to the differences in precipitation during the two cropping seasons. The precipitation was greater by 109.6% in the 2018 cropping season versus the 2017 cropping season (Fig. 1). Although N uptake was greater in 2017 than in 2018, similar trends were observed demonstrating that N uptake was greater in intercropped systems than in MM. Maize’s N uptake may be have been promoted in maize-legumes intercropping, and similar results may be seen even in the variable environment.

Root growth affects crop growth, and nutrient and water uptake. We found that the average biomass of maize roots was significantly increased by 52.6–60.4% in IMS and 64.7–82.3% in IMP compared with the MM over 2 years (Table 3). A well-developed fine root system replaced the large root biomass to enhance maize’s N uptake (Zeng & Peng, 2017). Maize roots showed an asymmetric horizontal distribution under IMS and IMP (Figs. 3B–3C, 3H–3I, 3E–3F and 3K–3L). However, a symmetric distribution of roots was observed in MM (Figs. 3A, 3G, 3D and 3J). The competitive use of nutrients and water between the component crops in the intercropping system was affected by the distribution of the roots ( Xia et al., 2013; Yong et al., 2015; Li et al., 2018). The competitive uptake of nutrients and water by the component crops altered the distribution of nutrients and water in the soil, thus, the crops regulated their root growth and spatial distribution to obtain the necessary nutrients and water for growth (Yu et al., 2014; Liu et al., 2020). We found that the intercropped maize’s roots extended into the soybean and peanut rows and into the rows between legumes (Figs. 3B–3C, 3H–3I, 3E–3F and 3K–3L). Our finding is consistent with the results of previous studies (Gao et al., 2010; Xia et al., 2013). Maize intercropped with legumes altered maize root distribution and increased their root absorption area. Intercropped maize’s total RSAD under the P2 significantly increased in IMS and IMP compared with MM (Fig. 2). Importantly, maize’s RSAD was greater in the interspecific rows between maize and legume than in intraspecific maize rows at most soil depths (Figs. 3B–3C, 3H–3I, 3E–3F and 3K–3L). The changes in the roots’ spatial distribution and maize’s increased root RSAD improved the aboveground N uptake in the intercropping system versus the MM (Tables 1 and 2). Maize’s total RSAD and root biomass were greater in IMP than in IMS (Fig. 3, Table 3), resulting in a higher N uptake in IMP than in IMS (Tables 1 and 2).

Our study confirmed that the root-bleeding intensity was closely related to active nutrient uptake in the root system. Sap bleeding reflects the roots’ physiological activity (Noguchi et al., 2005) and is affected by the environment and cultivation practices (Guan et al., 2014; Yang et al., 2016; Jia et al., 2018). The intensity of the root-bleeding significantly increased with intercropping and N application at the different growth stages for maize (Table 4). A previous study indicated a close relationship between bleeding intensity and root traits in maize (Morita et al., 2000). Maize intercropped with soybean and peanuts promoted root growth and changed the root distribution (Fig. 3 and Table 3), leading to a greater root-bleeding sap intensity in IMS and IMP than in MM at the different growth stages (Table 3). Our results suggested that maize intercropped with soybean and peanuts may enhance the physiological activity of maize roots, improving N uptake by the roots and aboveground N accumulation (Table 1). The heavy precipitation in 2018 affected the root-bleeding sap intensity and N uptake of maize when compared to 2017 (Fig. 1). The root-bleeding sap intensity and N uptake were significantly lower in 2018 than in 2017 (Table 1) but similar trends were observed showing that for maize roots bleeding was greater in intercropping than in MM. (Table 4). Sap’s nitrate-N content was significantly higher in intercropped systems compared with MM at the silking stage in 2018. The nitrate-N content significantly increased by 85.5–89.5% in IMS and by 91.2–102.4% in IMP compared with MM (Fig. 4). These results indicated that maize-legume intercropping could enhance root activity and increase the N uptake of maize roots.

Cultivation practices alter the soil environment to produce nutrient and water stress, and produce the ROS toxic effect (Hu et al., 2010; Liu & Jiang, 2017; Yao et al., 2019). The ROS (O2−, H2O2) are highly reactive and toxic, damaging DNA, proteins, liquids and carbohydrates to ultimately cause cell death (Gill & Tuteja, 2010) and accelerate crop roots senescence. To eliminate the excess ROS, antioxidant enzyme activities and contents, including superoxide dismutase (SOD), catalase (CAT) and glutathione (GSH), are increased (Gill & Tuteja, 2010). We determined the activities of the root SOD and CAT and found that the GSH content of maize were significantly increased in intercropping systems, compared with the MM (Fig. 5). Intercropped maize’s root SOD activity was significantly increased by 38.5–67.8% in IMS and 76.5–123.1% in IMP (Fig. 5A). Its CAT activity was significantly increased by 68.8–77.8% in IMS and 101.0–126.3% in IMP compared with MM (Fig. 5B). The GSH content was significantly increased by 15.7–36.4% in IMS and 19.7–45.8% in IMP compared with MM (Fig. 5C). These results suggest that intercropping regulated maize roots’ intracellular homeostasis, delayed the maize root senescence, and maintained roots’ nutrient acquisition by avoiding the redox reaction imbalance. Intercropping may increase the N uptake capacity and prolong the uptake time of the maize root system. Maize roots’ SOD and CAT activities were greater in IMP than in IMS (Figs. 5A and 5B). These results indicated that intercropped maize’s root senescence was slower in IMP than in IMS. Thus, the root absorptive capacity of intercropped maize was greater in IMP than in IMS (Fig. 4 and Table 3).

Conclusions

The maize-legume strip intercropping system significantly increased maize’s aboveground N uptake and N use efficiency compared with monocultured maize. Maize’s AE was greater in IMP than in ISM. Its RE was greater in IMS than in IMP. Maize roots extended under soybean and peanut roots and across the legume inter-rows in the intercropping system. Intercropping with soybean and peanuts significantly increased the RSAD and total root biomass of maize, which performed better in IMP than in IMS. Intercropping with soybean and peanuts increased the roots bleeding sap intensity, root antioxidant enzymes activity, and maize roots’ antioxidant content.

Our results suggest that maize intercropped with legumes can enhance the aboveground N uptake and N use efficiency of maize by promoting root growth, changing the spatial distribution of the roots, delaying root senescence, and improving root activity. Maize-legume strip intercropping may reduce the need for N fertilizer and improve N use efficiency.

Supplemental Information

Supplemental Information 1 N content of maize organs (g kg−1).

MM, monoculture maize; IMS, intercropped maize with soybean; IMP, intercropped maize with peanut; N0, no N fertilizer; N1, conventional N fertilizer.

Click here for additional data file.

Supplemental Information 2 Biomass of maize (g plant−1).

MM, monoculture maize; IMS, intercropped maize with soybean; IMP, intercropped maize with peanut; N0, no N fertilizer; N1, conventional N fertilizer.

Click here for additional data file.

Supplemental Information 3 Root biomass of maize (g plant−1 ).

MM, monoculture maize; IMS, intercropped maize with soybean; IMP, intercropped maize with peanut; N0, no N fertilizer; N1, conventional N fertilizer.

Click here for additional data file.

Supplemental Information 4 Root surface area of maize (cm−2 1570 cm−3).

MM, monoculture maize; IMS, intercropped maize with soybean; IMP, intercropped maize with peanut; N0, no N fertilizer; N1, conventional N fertilizer; P1, the inter-row of maize; P2, intra-row of maize; P3, the adjacent row of maize and legume; P4, intra-row of legume; P5, inter-row of legume.

Click here for additional data file.

Supplemental Information 5 Antioxidant enzyme activity and antioxidants content of maize root.

MM, monoculture maize; IMS, intercropped maize with soybean; IMP, intercropped maize with peanut; N0, no N fertilizer; N1, conventional N fertilizer.

Click here for additional data file.

Supplemental Information 6 Bleeding sap intensity of maize (g plant−1 12 h−1).

MM, monoculture maize; IMS, intercropped maize with soybean; IMP, intercropped maize with peanut; N0, no N fertilizer; N1, conventional N fertilizer; V12, the twelve-leaf-stage of maize; R1, the silking-stage of maize; R3, the milk-stage of maize.

Click here for additional data file.

Supplemental Information 7 Nitrate-N content in bleeding sap (μg g−1).

MM, monoculture maize; IMS, intercropped maize with soybean; IMP, intercropped maize with peanut; N0, no N fertilizer; N1, conventional N fertilizer.

Click here for additional data file.

Additional Information and Declarations

Competing Interests

Author Contributions

Data Availability

The authors declare that they have no competing interests.

Benchuan Zheng conceived and designed the experiments, performed the experiments, analyzed the data, prepared figures and/or tables, and approved the final draft.

Xiaona Zhang performed the experiments, analyzed the data, authored or reviewed drafts of the paper, and approved the final draft.

Ping Chen performed the experiments, analyzed the data, prepared figures and/or tables, and approved the final draft.

Qing Du performed the experiments, analyzed the data, authored or reviewed drafts of the paper, and approved the final draft.

Ying Zhou performed the experiments, analyzed the data, authored or reviewed drafts of the paper, and approved the final draft.

Huan Yang performed the experiments, analyzed the data, authored or reviewed drafts of the paper, and approved the final draft.

Xiaochun Wang conceived and designed the experiments, authored or reviewed drafts of the paper, and approved the final draft.

Feng Yang conceived and designed the experiments, authored or reviewed drafts of the paper, and approved the final draft.

Taiwen Yong conceived and designed the experiments, authored or reviewed drafts of the paper, and approved the final draft.

Wenyu Yang conceived and designed the experiments, authored or reviewed drafts of the paper, and approved the final draft.

The following information was supplied regarding data availability:

The raw data are available as a Supplemental File.

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
