# Peer review of "Improving maize’s N uptake and N use efficiency by strengthening roots’ absorption capacity when intercropped with legumes"

_PeerJ, doi:10.7717/peerj.11658_

## Round 0.1 · original submission · Minor Revisions

Please follow the notes from the 3 reviewers, enhance the discussion, and have the language revised by a fluent English speaker or a professional editing service.

Thank you,

Reviewer 1 ·

Basic reporting

Clear and unambiguous, professional English used throughout.
Literature references, sufficient field background/context provided.
Professional article structure, figures, tables. Raw data shared.
Self-contained with relevant results to hypotheses.

Experimental design

Methods described with sufficient detail & information to replicate.

Validity of the findings

All underlying data have been provided; they are robust, statistically sound, & controlled.
Conclusions are well stated, linked to original research question & limited to supporting results.

Additional comments

The maize intercropped with legumes for improving N uptake and N use efficiency by strengthening roots absorption capacity is extremely important, and the research idea is very interesting. The article is well written with very few mistakes found throughout, that should be resolved before it can be published.
Abstract section
Line 36: the sentence (The root-bleeding sap intensity of maize was significantly enhanced by 22.7%-49.3% in IMS and by 37.9%-66.7% in IMP, compared with the MM) not clear can rewrite it?
Introduction section
L78: Add reference for (When crop plants are suffered from environmental stress, the superoxide 88 anion radical (O2 - ) and hydrogen peroxide (H2O2) etc. are induced).
Line 91: Add reference for (Root antioxidation response can 92 delay the senescence of plant roots.)
Line 95: The authors said (There is a greater N uptake of maize when intercropped with legumes (Yong et al. 2012; Zhang et al. 2017)), but mentioned in line 72 (wheat-maize intercropping), Is it possible to mention about maize intercropped with legumes?
Material and methods section
Line 119: annual precipitation of 1009.4 mm. should it be mm3?
Line 144: please add comma (,) before and.
Line 158: The authors determined (SOD), (CAT), (GSH) contents. Please add more details about methods for readers.
Line 164: I think this sentence (Skimmed cotton was packed in a self-sealed bag and 165 placed over the plant stalk), it is duplicated. Please delete it.
Line 185 and 186: The authors wrote 2 equations by different type. Please use same size.
Line 188:(performedwith) should add space (performed with)
Line 189: replace first and with comma (,); Should be (SigmaPlot14.0, Origin 2017, and Surfer v. 8.0 190 were used to draw the figures).

Results section
Line 201-203: units (kg kg-1) should be superscript (-1). Please modify in whole manuscript.

Discussion section
Line 252: It will be better to write the name of crop for each citation; (Intercropping increased crops nutrients uptake is reported (Li et al. 2004; Li et al. 2003; Zhang et al. 2017)).
Line 302-303: Add citation please (Cultivation practices change the soil environment to produce nutrient and water stress, and produce the toxic effect of reactive oxygen species (ROS)).
The discussion contained a repetition for the results. I request the authors to reduce the results as much as possible in the discussion section.
The authors indicated In their study that loading of corn on peanuts or soybeans has enhanced the delaying the root senescence and improving root activity, without discussing the mechanism of this enhancement.
Conclusion section
Recommendations for future studies are needed in the Conclusion chapter. Kindly provide strong recommendations for future researches.

References section:
Line 438: The reference missing some details (Yao, Y., Zhang, C., Camberato, J. J., & Jiang, Y. (2019). Nitrogen and carbon contents, nitrogen use efficiency, and antioxidant responses of perennial ryegrass accessions to nitrogen deficiency. Journal of Plant Nutrition, 42(17), 2092-2101.)

Hopefully, my comments improve the manuscript.

Reviewer 2 ·

Basic reporting

English should be corrected, that the content was more clear and grammatically correct.
The article includes sufficient introduction and background.
Relevant literature and appropriately referenced are used.

Experimental design

Methods should be described in more detail.
There is insufficient information on the determination of Root surface area density (RSAD).
Statistical information should be described in more detail.

Validity of the findings

In the manuscript the authors determined the effect of different maize-legume strip intercropping system on underground root growth, N uptake and N use efficiency of maize.

Additional comments

The manuscript shows data on weather conditions. For what purpose? Years of study were varied in terms of rainfall. What was the influence of years on the features of the roots (e.g. RSAD) and other parameters? Does improving the N uptake and nitrogen efficiency of maize by strengthening roots absorption capacity when strip intercropped with legumes depend on weather conditions?
Fertilization rates should be presented in the pure component.
The authors included too much detailed information about the results in the Abstract and Conclusions. In conclusions, general statements should be included.

Reviewer 3 ·

Basic reporting

English throughout the work should be checked and corrected, that the content was clear and grammatically correct.
Originality/novelty is average.
Quality of presentation is average.
Literature well referenced and relevant.
Structure conforms to PeerJ standards.
Figures are relevant, well labelled and described.

Experimental design

The methods should be more adequately described.
The descriptions in the methodology should be more precise.

Validity of the findings

Authors gives some new information about the intercropped cultivation of maize with legumes

Additional comments

The introduction provide sufficient background and include all relevant references.
The results are rather clearly presented and the conclusions are supported by the results.
But the methods should be more adequately described.The descriptions in the methodology should be more precise.

English throughout the work should be checked and corrected, that the content was clear and grammatically correct.
Information on technical and other errors is provided in the text.

Specific comments
Line 39-40 - uncommon abbreviations should be spelled out at first use: GSH, SOD, CAT

Line 102-105 - Comment on language and grammar issues - the English language should be improved to ensure that an international audience can clearly understand your text. Some examples where the language could be improved “Additionally, there is limited information on the intercropping delaying maize root senescence, and how the change of antioxidant enzyme activity and antioxidants in the root system of intercropping maize is still unknown”.

Line 139-143 - the dates of sowing and harvesting individual plants in the years of use are not clearly indicated (too many numbers)

Line 152-153 - the company and country of production should be specified

Line 157 - Then, some were dried at 85 ℃ to determine the biomass. It is not clear. If mass was determined, why was only "some"

Line 188-190 - Statistical analysis- should be described in more detail. What Anova was used? Was the same method used to evaluate all the features?

Line 324-326- Conclusions - in conclusions, the exact values of the assessed features should not be quoted, but general statements should be made

The weather conditions are given in Figure 1, but no description is provided. The years of research were very varied in terms of precipitation. In one year, rainfall is probably 50% less than in the second. The amount of precipitation in individual years is not given. The influence of weather conditions (precipitation) on the obtained results is also not taken into account.

Other information is provided in the text.

Annotated reviews are not available for download in order to protect the identity of reviewers who chose to remain anonymous.

---

## Round 0.2 · accepted · Accept

The manuscript is ready for publishing.

Reviewer 1 ·

Basic reporting

No comments

Experimental design

No comments

Validity of the findings

No comments